# Role of Melatonin in the Onset of Metabolic Syndrome in Women

**DOI:** 10.3390/biomedicines11061580

**Published:** 2023-05-30

**Authors:** Vania Miloucheva Peneva, Dora Dimitrova Terzieva, Mitko Dimitrov Mitkov

**Affiliations:** 1Department of Clinical Laboratory, Faculty of Pharmacy, Medical University, 4002 Plovdiv, Bulgaria; terzieva2006@yahoo.com; 2Department of Endocrinology, Faculty of Medicine, Medical University, 4002 Plovdiv, Bulgaria; mitko.mitkov@mu-plovdiv.bg

**Keywords:** metabolic syndrome, women, melatonin, leptin, ghrelin

## Abstract

Metabolic syndrome (MetS) is a constellation of several associated cardiometabolic risk factors that increase the risk of developing type 2 diabetes mellitus (T2DM), cardiovascular diseases, and mortality. The role of hormonal factors in the development of MetS is assumed. In women, an insulin-resistant state that is associated with polycystic ovarian syndrome and increased deposition of intra-abdominal adipose tissue promotes the development of MetS and increases cardiovascular risk. The neuroendocrine hormone melatonin is secreted mainly at night under the regulatory action of the suprachiasmatic nucleus in the hypothalamus. Melatonin secretion is influenced by exogenous factors such as light and seasons and endogenous factors such as age, sex, and body weight. At present, the role of melatonin in metabolic disorders in humans is not fully understood. In this review, we set out to analyze the relationship of melatonin with the main features of MetS in women. Data from experimental and clinical studies on the role of melatonin in glucose metabolism and on the involvement of melatonin in lipid disturbances in MetS are reviewed. The complex influence of melatonin on hypertension is discussed. The changes in melatonin, leptin, and ghrelin and their relation to various metabolic processes and vascular dysfunction are discussed.

## 1. Introduction

According to the modern concepts, MetS is a constellation of several inter-related cardiometabolic risk factors such as abdominal obesity, hyperglycemia, atherogenic dyslipidemia, increased blood pressure, and prothrombotic and proinflammatory conditions. These factors increase the risk of developing type 2 diabetes, cardiovascular disease, and mortality [1]. The role of hormonal factors in the development of MetS is discussed in the scientific literature [2]. There are data on the involvement of the hypothalamic–pituitary–adrenal axis in the pathogenesis of MetS [3,4]. It is believed that there are ethnic, racial, and gender differences in the prevalence and characteristics of MetS between women and men [5,6,7]. Compared to men, in women, the main characteristics of MetS (abdominal obesity, dyslipidemia, hyperglycemia, hypertension) are so altered that they lead to higher morbidity and mortality from cardiovascular diseases [5]. MetS in women has some unique features [5,7]. For example, according to Bentley-Lewis et al. [5], unique modifiers of the MetS in women are pregnancy, lactation, gestational diabetes mellitus, pre-eclampsia, polycystic ovary syndrome, hormonal contraceptive treatment, and menopause. An insulin-resistant state associated with polycystic ovary syndrome and increased deposition of intra-abdominal adipose tissue promotes the development of MetS and increases cardiovascular risk [7].

There are data that show that about 47 million US residents live with MetS, and the prevalence of MetS differed slightly among men (24%) and women (23.4%) [8]. When the data reported in The Third National Health and Nutrition Examination Survey (NHANES) III for the period 1988–1994 are compared with the data from NHANES 1999 to 2000, the age-adjusted prevalence of MetS increases by 23.5% among women (*p* = 0.021) and 2.2% among men (*p* = 0.0831), and it is mainly due to hypertriglyceridemia and high blood pressure among women [9]. Studies indicate controversial data on the prevalence of MetS among both sexes. Some researchers have shown a comparable involvement of the sexes [8,9,10], others have indicated predominance of the syndrome in females [11,12,13], and still others have indicated males as the more affected sex [14]. Gender-specific differences in characteristics are not consistent across studies. Among 103,763 participants with MetS, Lee et al. [15] found a gender difference in the components of Mets expressed in both sexes. In females, the most commonly seen components are elevated triglycerides (79.0%), followed by blood sugar (78.6%) and high blood pressure (78.5%). In males, the most commonly observed MetS components are blood sugar (87.5%), triglycerides (83.5%), and blood pressure (83.1%). Among females, 10.4% had five components of MetS, in contrast to 5.2% of males. Ogbera [10] present similar distribution of MetS in both sexes (males 83%, females 86%), except for ages 70–79, for which the proportion of males with MetS is almost twice that of females. Other researchers such as Beigh et al. [11] have indicated a prevalence of MetS in women compared to men (29% vs. 23%), and they have also observed gender differences in MetS components, with a prevalence in the women’s group of high waist circumference 63% and hyperglycemia 42%. In Chinese adults (*n* = 2334), He, Y. et al. [12] also reported a higher prevalence of MetS in women compared to men (54.1% vs. 34.8%). Threefold greater odds of having metabolic syndrome for women compared to men are reported in the study of Njelekela et al. [13]. Other researchers report a higher incidence of MetS in men [14]. The prevalence of MetS in obese males is greater than in females (67.6% vs. 45.0%), and this difference is even higher for men < 40 years (72.5% vs. 36.8%). The summarized data of gender differences in the prevalence and characteristics are presented in Table 1.

Melatonin is a neuroendocrine hormone that is synthesized in the principal cells of the pineal gland, namely, pinealocytes, from the amino acid tryptophan [16]. This process is multi-step and enzyme-catalyzed, involving the enzymes L-triptophan decarboxylase, arylalkylamine N-acetyltransferase (AANAT), tryptophan hydroxylase, and hydroxy-indole-O-methyl-transferase (HIOMT) [17,18]. AANAT and HIOMT are the two essential enzymes that regulate the rate of melatonin synthesis. Once synthesized, melatonin is not stored in the pineal gland. In the blood, about 70% of melatonin is bound to albumin, and the remaining proportion of about 30% is taken up by the surrounding tissues. After melatonin is hydroxylated in the liver, it is excreted in the urine. The biological action of melatonin in humans is mediated by two G-protein-coupled receptors (MT_1_ and MT_2_), one cytoplasmic quinone reductase receptor (MT_3_), and one nuclear retinoid-related orphan receptor (RZR/RORα) [19,20,21]. MT_1_ is encoded on chromosome 4, contains 351 amino acids, and inhibits adenylate cyclase, while MT_2_ is encoded on chromosome 11 and consists of 363 amino acids, inhibiting adenylate cyclase and guanylyl cyclase. Melatonin receptors MT_1_ and MT_2_ are widely represented in the pituitary gland, in structures of the central nervous system (hypothalamus, suprachiasmatic nucleus, hippocampus, paraventricular nuclei, retina), in the cardiovascular system (heart chamber wall, aorta, coronary and cerebral arteries), in the gastrointestinal tract (the mucosa of the colon and jejunum, appendix), in the digestive system (gall bladder, liver), and in various cells (of the immune system, platelets, epithelial cells of the prostate and breast, granulosa cells of the ovaries, myometrium, placenta, and kidney of the fetus, brown and white adipocytes, etc.) [20,21]. MT_3_ is localized in the liver, kidney, heart, lung, small intestine, muscle, and brown adipose tissue and has predominantly antioxidant effects [21]. The two described isoforms (RORα1 and RORα2) of the retinoid-related orphan nuclear receptor mediate melatonin binding to nuclear transcription factors. The melatonin-related orphan receptor X (ML_1_X) is also known, which probably supports melatonin binding to MT_1_.

Melatonin is secreted mainly at night under the regulatory action of the suprachiasmatic nucleus [22,23]. The disturbed function of the circadian clock leads to changes in several physiological cycles (sleep/wake cycle, body temperature cycle) and the rhythmic secretion of hormones such as insulin, leptin, and cortisol [19]. There is increasing evidence of the harmful influence of disturbed circadian organization on metabolism [24]. Melatonin, through its pronounced circadian secretion rhythm, is thought to act as an internal synchronizer of the circadian system [25]. Perhaps that is why some researchers call it “a biological modulator” of circadian rhythms [26]. Reduced or absent melatonin secretion can lead to development of insulin resistance, glucose intolerance, insulin secretion disorders, energy balance disorders, and obesity [27].

In the present literature review, we aimed to analyze the inter-relationship of melatonin with some MetS risk factors such as dysglycemia, dyslipidemia, and hypertension in women. Changes in melatonin, leptin, and ghrelin concentration and their relationship with various metabolic processes and vascular dysfunction are discussed. Data for the review were collected by using Scopus, PubMed, ScienceDirect, and Google Scholar databases, published between 2000 and 2022. The following keywords were used: “metabolic syndrome”, “melatonin”, “women”, “ghrelin”, and “leptin”. In total, 254 studies were found, then 62 of them were selected for this systematic review.

## 2. Role of Melatonin in Carbohydrate Metabolism

It is believed that the disturbed rhythm of melatonin secretion leads to changes in carbohydrate metabolism, affecting the secretory activity of pancreatic islet cells, glucose metabolism in the liver, and insulin sensitivity of target tissues [28]. According to an experimental study by Nduhirabandi et al. from 2014 [29], in obese rats, melatonin administration for 3 or 6 weeks reduced serum insulin concentration and Homeostatic model assessment (HOMA) index values. It did not affect weight gain, visceral adiposity, serum triglycerides, and glucose. According to a recent study by Nduhirabandi et al. [30], short-term melatonin administration in vivo improves basal glucose uptake and insulin response in insulin-resistant cardiomyocytes isolated from obese experimental models.

In the last 10–15 years, researchers have noticed an increased interest in studying the role of melatonin in glucose metabolism as a risk factor for the development of T2 DM or in its treatment in humans [31]. Despite intensive studies in this direction, there are some contradictions in the data on the influence of melatonin on carbohydrate metabolism. Whether melatonin increases or decreases fasting blood glucose, decreases glucose tolerance, or increases the risk of developing T2DM remains to be elucidated. A number of studies have shown a negative correlation between endogenous nocturnal melatonin and the risk of developing insulin resistance. In a case–control study, McMullan et al. [32] analyzed the relationship between melatonin secretion and the risk of developing T2DM in nurses by examining the nocturnal excretion of the major melatonin metabolite 6-sulfatoxymelatonin. They found that low melatonin secretion was associated with an increased risk of developing T2DM. According to another study by McMullan et al. [33], in non-obese women without T2DM, a slight decrease in nocturnal melatonin excretion was associated with increased insulin resistance. It is also established that low nocturnal secretion of 6-sulfatoxymelatonin inversely correlates with insulin resistance and insulin values in individuals with prediabetes [24]. However, Rubio-Sastre et al. found, in a study with 21 healthy women (body mass index 23.0 ± 3.3 kg/m^2^), that melatonin administration impaired glucose tolerance, regardless of the timing of its administration [34]. An opinion has been expressed that evening melatonin probably impairs glucose tolerance primarily by reducing insulin sensitivity, concomitant with insufficient beta-cell compensation, and morning melatonin suppresses insulin release. The randomized placebo-controlled trial of Terry et al. also showed that melatonin supplementation of 8 mg/day for 10 weeks versus placebo moderately improved some metabolic components such as waist circumference (*p* = 0.15), triglycerides (*p* = 0.17), HDL cholesterol (*p* = 0.59), and systolic blood pressure (*p* = 0.013) but resulted in a slight, non-significant worsening in fasting plasma glucose (*p* = 0.29) [35,36].

An interesting relationship was established between a mutation in the gene encoding the melatonin receptor MT_2_ (MTNR1B) and glucose tolerance. In healthy female carriers of the mutation, melatonin administration decreased glucose tolerance [37]. The effect of melatonin on glucose concentration at 120 minutes during the oral glucose tolerance test was significantly different when compared between carriers and non-carriers of the mutation. Wei et al. [38] also investigated the relationship between the rs10830963 variant in the MTNR1B gene and the development of gestational diabetes mellitus (GDM) and the effect of the MT_2_ receptor on glucose uptake and transport in the trophoblast. The study revealed that this variant is significantly associated with GDM, elevated fasting plasma glucose, and 1 h and 2 h plasma glucose. In a review article, Verteramo et al. [39] tried to evaluate the role of melatonin during normal pregnancy, in high-risk pregnancy (GDM, pre-eclampsia, intrauterine growth restriction), during birth, and in the first period after delivery. They stated also that MT_2_ gen polymorphisms positively correlated with an increased risk of glycemic disorders.

## 3. Role of Melatonin in Lipid Metabolism

In addition to maintaining carbohydrate metabolism, it is believed that melatonin is likely to be an essential factor in lipid regulation. A number of experimental and clinical studies support the beneficial effect of melatonin on lipid profile parameters. In pinealectomized rats, there was a significant increase in the plasma concentrations of total cholesterol, triglycerides, low-density lipoprotein cholesterol (LDL-cholesterol), and very-low-density lipoprotein cholesterol (VLDL-cholesterol), and a decrease in high-density lipoprotein cholesterol (HDL-cholesterol), and the administration of melatonin (5 mg/kg for 28 days) improved the concentrations of these indicators [40]. Other experimental studies have reported similar data on the beneficial effect of melatonin intake on total cholesterol and LDL-cholesterol [41]. Al-Mhbashy et al. [42] aimed to evaluate the effect of melatonin on oxidative stress, glycated hemoglobin (HbA_1_c), microalbuminuria, and lipid profile in 30 patients with T2DM who received oral melatonin 3 mg/day for 3 months. They found that elevated total serum cholesterol decreased significantly after the 60th day from the start of therapy (20%, *p* < 0.05), and after 90 days of treatment, it reached values lower than those of healthy controls; LDL-cholesterol also decreased on day 60 (36%, *p* < 0.05) and reached a 46% decrease after day 90; triglycerides showed a 22% decrease after day 90, which was non-significant compared to baseline values. They found a significant increase in HDL-cholesterol on the 30th day from the start of therapy (15%, *p* < 0.05); on the 90th day, the increase was 30%. Plasma malondialdehyde (as an indicator of lipid peroxidation) decreased significantly after the 60th day of melatonin administration, and glucose and HbA_1_c decreased significantly after the 90th day. In a study in Iraq, a double-blind, placebo-controlled trial was conducted on 20 women with MetS treated with metformin and melatonin (10 mg daily at bedtime) for 3 months and 15 women with MetS treated with metformin and placebo [43]. Based on the data obtained, the researchers conclude that melatonin improves the effect of metformin on several components of the syndrome (fasting serum glucose, lipid profile, body mass index (BMI), insulin resistance, hyperinsulinemia) compared to treatment with metformin alone.

## 4. Effect of Melatonin on Blood Pressure

Currently, the mechanisms by which melatonin affects blood pressure are thought to be complex and incompletely understood [44]. In this regard, the following are discussed: the receptor-mediated and receptor-independent effects of melatonin on blood pressure, the effect of melatonin on arterial blood pressure by influencing the activity of the sympathetic nervous system or by influencing structures of the central nervous system, and the potential role of melatonin in the regulation of nocturnal blood pressure values. Forman et al. [45] analyzed the relationship between 6-sulfatoxymelatonin excretion in first-morning urine and the risk of hypertensive episodes in young pre-menopausal women (*n* = 554) aged 25 to 42 years who were followed for 8 years. Women with low morning melatonin excretion (*n* = 125) were found to have hypertensive episodes. It is hypothesized that low nocturnal melatonin concentrations may be a pathophysiological factor in the development of hypertension. A recent clinical study compared salivary melatonin at different times of the day (morning between 06:00 and 08:00, mid-day between 11:00 and 13:00, evening between 18:00 and 20:00 hours and before sleep) in pregnant women (*n* = 58) with hypertension or impaired glucose metabolism with that of healthy pregnant women (*n* = 40), also assessing sleep quality and sleep–wake cycle [46]. Women with pregnancy complications were found to have smaller circadian variation in melatonin secretion, with their daily melatonin concentrations being lower than those of healthy pregnant women. Pregnant women with these complications have shorter nocturnal sleep times. Another study by Cai et al. [47] from 2020 compared endogenous daytime plasma melatonin in both untreated pulmonary hypertension patients (*n* = 64, 53% women) and healthy controls (*n* = 111. 64% women) and experimental models with and without pulmonary hypertension. No statistically significant sex-related difference in melatonin was found in controls and patients with untreated pulmonary hypertension. Compared with healthy controls, untreated patients with pulmonary hypertension have elevated diurnal plasma melatonin. The pathophysiological mechanism of this increase is currently unclear. According to data from this study, high melatonin concentrations are an independent risk factor for pulmonary hypertension, and low melatonin is associated with poor long-term patient survival. A strong correlation was found between melatonin and heart rate in subjects with pulmonary hypertension, whereas in controls, melatonin correlated with systolic blood pressure.

Several scientific studies have analyzed the influence of exogenous melatonin on systolic and diastolic blood pressure values. In a placebo-controlled study, 18 women (9 had treated essential hypertension, whereas 9 were normotensive) received slow-release melatonin 3 mg/day for 3 weeks or a placebo [48]. The results show that long-acting melatonin intake does not affect the daily values of blood pressure or heart rate. A significant decrease in systolic (*p* = 0.0423), diastolic (*p* = 0.0153), and mean arterial pressure (*p* = 0.013) was found at night, resulting in an improved day–night blood pressure rhythm. Możdżan et al. [49] evaluated the effect of melatonin on blood pressure in patients with treated essential hypertension and T2DM with reasonable metabolic control. Patients are divided into two groups depending on the degree of drop in blood pressure at night: the so-called “non-dipper” group (with a physiological drop in blood pressure of less than 10%) and a dipper group (with a physiological drop in blood pressure of 10–20%). Both groups received 4 weeks of 3 mg/day melatonin and another 4 weeks of 5 mg/day. The study data showed that over 30% of the non-dipper group with T2DM treated with melatonin restored the normal circadian rhythm of blood pressure, affecting nocturnal systolic, diastolic, and mean arterial blood pressure. These doses of melatonin did not reduce the value of blood pressure in the dippers with T2DM. Based on meta-analysis data, it is established that evening administration of long-acting melatonin has an antihypertensive effect by affecting nocturnal blood pressure values [50]. A commentary by Simko et al. of 2019 [51] discusses the possibility of melatonin being used as a rational alternative to the conservative treatment of resistant hypertension.

## 5. Relationship between Melatonin, Leptin, and Ghrelin

The development of MetS and subsequent adverse complications directs researchers to search for new risk factors [52]. A certain amount of data shows that hormones regulating the body’s energy balance are potential risk factors for MetS. The complex interplay between metabolic and hormonal factors in women with MetS is evidenced by several studies, according to which, hormonal indicators such as melatonin, ghrelin, and leptin correlate with disturbances in glucose homeostasis and changes in body weight. Leptin is a key signaling molecule in the regulation of energy homeostasis and food intake and probably plays a role as a metabolic and neuroendocrine hormone [53]. Hyperleptinemia and leptin resistance are closely related to obesity and T2DM. A significant positive correlation of leptin with BMI, waist circumference, body fat, insulin, insulin resistance, total cholesterol, and HOMA-index was found in 107 women aged 67 to 78 years [54]. A study by van Zyl et al. [55] analyzed the relationship between anthropometric and metabolic parameters, leptin, and adiponectin in 135 African women with MetS of various BMI, aged 26 to 63 years, living in an urban environment. It has been reported that as BMI increases, adiponectin decreases, and leptin and the leptin/adiponectin ratio increase. Leptin was positively correlated with BMI and waist circumference, the leptin/adiponectin ratio was positively correlated with fasting glucose, and adiponectin was negatively correlated with BMI and blood glucose.

According to some studies, night-shift work affects circadian rhythms and is a risk factor for obesity and a number of chronic diseases [56]. They found that healthy premenopausal nurses working the night shift had significantly lower morning melatonin and non-significantly lower leptin and higher ghrelin, BMI, HOMA-index, and biochemical indicators (serum glucose, insulin, HDL-cholesterol, LDL-cholesterol, triglycerides) compared to these parameters in nurses working the day shift. A positive correlation was found between HDL-cholesterol and melatonin and a negative correlation between insulin and leptin in night-shift workers, while in day-shift workers, melatonin correlated negatively with insulin and HOMA-index. They conclude that low melatonin and a disturbed balance between leptin and ghrelin contribute to the development of MetS.

Adipose tissue produces various proinflammatory cytokines (tumor necrosis factor-ɑ, interleukin-6 (IL-6)) and adipokines (leptin and adiponectin), which induce the enhanced formation of reactive oxygen species and oxidative stress [57]. In this regard, the effect of melatonin supplementation (6 mg/day for 40 days) on indicators of oxidative stress and inflammation were studied in obese women with a BMI ≥ 30 [58]. A significant decrease in mean serum concentrations of tumor necrosis factor-ɑ, interleukin-6, high-sensitivity C-reactive protein, and malondialdehyde was found in the group of obese women. These data support the positive effect of melatonin intake in obese women, which could be administered with an appropriate dietary regimen and physical activity.

A study in Brazil investigated whether a combination of gonadotropins, sex hormones, metabolic risk factors, and inflammatory markers could distinguish obese from non-obese women of different menopausal statuses [59]. It was established that postmenopausal women had higher fasting serum glucose and leptin than premenopausal women. BMI positively correlated with IL-6 level and negatively with follicle-stimulating hormone in postmenopausal women. It correlated positively with leptin in premenopausal women. By discriminant analysis, the combination of follicle-stimulating hormone and leptin was found to be an independent predictor of obesity in pre- and postmenopausal women. Jabbari et al. [52] investigated the relationship between serum ghrelin concentrations and brain-derived neurotrophic factors in premenopausal women with MetS. They found that compared to healthy controls (*n* = 43), women with MetS (*n* = 43) had statistically significantly lower serum concentrations for both proteins, which correlated negatively with triglycerides, fasting glucose, and HOMA-index and positively with HDL-cholesterol. The reason for the low ghrelin concentrations in women with MetS is not clear. According to one of the stated hypotheses, hyperinsulinemia and hyperglycemia probably suppress the secretion of ghrelin, but for now, the exact mechanism remains to be clarified.

The potential association of total, acylated, and non-acylated ghrelin with carotid artery atherosclerosis and gender in MetS patients was analyzed in a study by Zannetti et al. [60]. Gender differences were found in the correlation of ghrelin with subclinical carotid artery atherosclerosis. Total and non-acylated ghrelin were significantly higher in women compared to men. An association between non-acylated ghrelin and thickening of the intima and media of the carotid artery wall was found only in women with MetS, suggesting a sex-specific influence of non-acylated ghrelin in the development of atherosclerosis. Data from a more recent experimental study by Zannetti et al. [61] confirmed the beneficial effect of non-acylated ghrelin on vascular dysfunction associated with obesity and MetS. It lowers vascular oxidative stress, which protects against early atherosclerosis and the accumulation of lipids in the vascular wall. 

## 6. Conclusions

Metabolic syndrome is a significant risk factor for the development of cardiovascular disease and diabetes. The association of melatonin with several classic syndrome features (carbohydrate and lipid disorders, hypertension) in women has been discussed in experimental and clinical studies. In recent years, more and more attention has been paid to the role of some new additional risk factors for the development of MetS, such as the hormones melatonin, leptin, and ghrelin. Clarifying the relationship between the primary and additional features of the syndrome in women is essential for the complex etiopathogenetic evaluation of MetS. The data given in this review can help to achieve a better understanding of the differences in the prevalence and components of MetS in women and to look at integrating them into appropriate guidelines for a future complex therapeutic approach toward the components of the syndrome in affected individuals.

## Figures and Tables

**Table 1 biomedicines-11-01580-t001:** Gender differences in the prevalence and characteristics of MetS.

Authors/Refs	Study Design	All Participants/%Mets Males/%MetS Females/%MetS	Ages	Country/Region	Diagnostic Criteria	MetS Characteristics Males	MetS Characteristics Females
Ford, E.S. et al. [8]	Cross-sectional health survey	8814/23.7 4265/24 4549/23.4	≥20	US	NCEP ATPIII	↑Abd Ob 29.8% ↑TG 35.1% ↑BP 38.2% ↑FG 15.6%	↑Abd Ob 46.3% ↑TG 24.7% ↑BP 29.3% ↑FG 10.0%
Ford, E.S. et al. [9]	Cross-sectional health survey	1677/27 841 836	≥20	US	NCEP ATPIII	↑Abd Ob 36% ↑BP 40.9% ↑TG 35.6% ↑FG 37.7% ↓HDL C 36.6%	↑Abd Ob 51.9 ↑BP 37.3% ↑TG 29.9% ↑FG 23.8% ↓HDL-C 43.4%
Lee, S. et al. [15]	Nationwide cross-sectional survey	103,763/100 43,129/100 60,634/100	66	Korean	AHA/NHLBI, WHO for obesity in Asian–Pacific region	↑FG 87.5% ↑TG 83.5% ↑BP 83.1%	↑FG 78.6% ↑TG 79.0% ↑BP 78.5%
Ogbera, A.O. [10]	Cross-sectional survey	963/86 703/83 260/86	35–85	Nigeria	IDF TF	↓HDL-C ↑TG	↑BMI ↑HBA1c ↑LDL-C ↑TChol
Beigh, S.H. et al. [11]	Comparative study	500/25.6 294/23 206/29	≥30	India	NCEP ATP III Asian modified	↑WC42% ↑FG 25% ↓HDL-34%	↑WC 63% ↑FG 42% ↓HDL-C25%
He, Y. et al. [12]	Population-based cross-sectional study	2334/46.3 943/34.8 1391/54.1	≥60	China	IDF2005	↓HDL C 25.5% ↑TG 31.5% ↑BP 88%	↓HDL-C 41.9% ↑TG 43.7% ↑BP 82.6%
Njelekela, A.M. et al. [13]	Cross-sectional epidemiological study	209/38 115/23 94/53	44–66	Africa	NCEP ATPIII	↑BMI 13% ↑WC 11% ↑WHR 51% ↑BP	↑BMI 35% ↑WC 58% ↑WHR 73% ↓HDL-C ↑FG
Strack, C. et al. [14]	Cross-sectional study	432 173/67.6 259/45	18–69	Germany	NCEP ATP III	↑WC ↑FG ↑TG	↑HDL-C

↑: increased; ↓: decreased; Male: (M); Female: (F); WC: waist circumference; BMI: body mass index; WHR: Waist-to-height ratio; AHA/NHLBI: American Heart Association/National Heart, Lung, and Blood Institute; IDF: International Diabetes Federation; NCEP ATP III: Third Report of the National Cholesterol Education Program Expert Panel of Detection, Evaluation, and Treatment of High Blood Cholesterol in Adults; BP: blood pressure; TG: triglycerides; FG: fasting glucose; HBA1c: glycosylated hemoglobin level; LDL-C: low-density lipoprotein cholesterol; TChol.: total cholesterol.

## Data Availability

The data used in this article are sourced from materials mentioned in the references section.

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
