# Peer review of "Role of Melatonin in the Onset of Metabolic Syndrome in Women"

_biomedicines, 2023, doi:10.3390/biomedicines11061580_

Round 1

Reviewer 1 Report

Overall, this is an intersesting review publication and offers a good overview on the role of melatonin in female metabolic syndrome. However, before publication some points need to be clarified.

My comments:

Line 54 – the pituitary gland is part of the endocrine system but not CNS.

Line 56 – Neither the liver nor gall bladder are part of gastrointestinal tract. Both organs belong to digestive system but not GIT.

Line 75 – please add a brief methodology of this review.

Line 75 – please also indicate more what is novelty of this study.

Line 90, 98 and throughout the text – type 2 diabetes mellitus has been already abbreviated to T2DM (line 12). Please use consequently once introduced acronyms.

Line 111 – please decide between “MT2” or “MT2” (line 48).

Line 136 – Does country localization affect the obtained results? If yes, please discus how.

Line 190 – how “the blood pressure can be improved”?

Line 260 – please add future perspectives in the field.

Author Response

Dear reviewer , as an attachment I am sending the corrections made by me , according to your recommendations.

I remain at your disposal

Sincerely

Dr. Vanya Peneva

Reviewer 2 Report

In this review article, the authors tried to review and analyze the interrelationship of melatonin with some metabolic syndrome (MetS) risk factors, such as dysglycemia, dyslipidemia, and hypertension in women.

Comments

The reviewer has some concerns as follows:

1. Why choose women as the main axis of the discussion? The authors mentioned that MetS in women has some unique features (ref. 7 in this manuscript). Bentley-Lewis et al. have shown that the metabolic syndrome is estimated to be present in 47 million US residents with a similar age-adjusted prevalence in men (24%) and women (23%) (https://doi.org/10.1038/ncpendmet0616). Gender differences in metabolic disease have been studied. Moreover, a randomized controlled trial of melatonin supplementation in men and women with the metabolic syndrome has been done (https://doi.org/10.2147/OAJCT.S39551). A comparison of women to men should make this review article more interesting.

2. The presentation of Tables or Figures can enhance the reader's understanding for this review article.

3. In the Conclusions section, does it refer to the condition of women? Because it is not emphasized that it is under woman condition. However, it is still suggested that the conditions of women and men should be compared.

Author Response

Dear reviewer, I am sending you my corrections.

I sincerely hope for your permission

Peneva V.

Round 2

Reviewer 2 Report

This revised manuscript can be accepted. No further comments.